# Subquadratic Kronecker Regression with Applications to Tensor Decomposition

**Matthew Fahrbach**[*]
Google Research
fahrbach@google.com

**Gang Fu**
Google Research
thomasfu@google.com

**Mehrdad Ghadiri**
Georgia Tech
ghadiri@gatech.edu

## Abstract

Kronecker regression is a highly-structured least squares problem $\min_{\mathbf{x}} \|\mathbf{K}\mathbf{x} - \mathbf{b}\|_2^2$, where the design matrix $\mathbf{K} = \mathbf{A}^{(1)} \otimes \cdots \otimes \mathbf{A}^{(N)}$ is a Kronecker product of factor matrices. This regression problem arises in each step of the widely-used alternating least squares (ALS) algorithm for computing the Tucker decomposition of a tensor. We present the first *subquadratic-time* algorithm for solving Kronecker regression to a $(1 + \varepsilon)$-approximation that avoids the exponential term $O(\varepsilon^{-N})$ in the running time. Our techniques combine leverage score sampling and iterative methods. By extending our approach to block-design matrices where one block is a Kronecker product, we also achieve subquadratic-time algorithms for (1) Kronecker ridge regression and (2) updating the factor matrices of a Tucker decomposition in ALS, which is not a pure Kronecker regression problem, thereby improving the running time of all steps of Tucker ALS. We demonstrate the speed and accuracy of this Kronecker regression algorithm on synthetic data and real-world image tensors.

## 1 Introduction

Tensor decomposition has a rich multidisciplinary history with countless applications in data mining, machine learning, and signal processing [35, 55, 58, 31]. The most widely-used tensor decompositions are the CP decomposition and the Tucker decomposition. Similar to the singular value decomposition of a matrix, both decompositions have natural analogs of *low-rank* structure. Unlike matrix factorization, however, computing the rank of a tensor and the best rank-one tensor are NP-hard [27]. Therefore, most low-rank tensor decomposition algorithms decide on the rank structure in advance, and then optimize the variables of the decomposition to fit the data. While conceptually simple, this approach is extremely effective in practice for many applications.

The *alternating least squares* (ALS) algorithm is the main workhorse for low-rank tensor decomposition, e.g., it is the first algorithm mentioned in the MATLAB Tensor Toolbox [7]. For both CP and Tucker decompositions, ALS cyclically optimizes disjoint blocks of variables while keeping all others fixed. As the name suggests, each step solves a linear regression problem. The *core tensor* update step in ALS for Tucker decompositions is notoriously expensive but highly structured. In fact, the design matrix of this regression problem is the Kronecker product of the factor matrices of the Tucker decomposition $\mathbf{K} = \mathbf{A}^{(1)} \otimes \cdots \otimes \mathbf{A}^{(N)}$. Our work builds on a line of Kronecker regression algorithms [17, 18, 47] to give the first *subquadratic-time* algorithm for solving Kronecker regression to a $(1 + \varepsilon)$-approximation while avoiding an exponential term of $O(\varepsilon^{-N})$ in the running time.

We combine leverage score sampling, iterative methods, and a novel way of multiplying sparsified Kronecker product matrices to fully exploit the Kronecker structure of the design matrix. We also extend our approach to block-design matrices where one block is a Kronecker product, achieving

---

[*]Authors are listed alphabetically. A preliminary version of this work that focuses on efficient sketching for Tucker decompositions appears in `arXiv:2107.10654` [21].

36th Conference on Neural Information Processing Systems (NeurIPS 2022).

subquadratic-time algorithms for (1) Kronecker ridge regression and (2) updating the factor matrix of a Tucker decomposition in ALS, which is not a pure Kronecker regression problem. Putting everything together, this work improves the running time of all steps of ALS for Tucker decompositions and runs in time that is sublinear in the size of the input tensor, linear in the error parameter $\varepsilon^{-1}$, and subquadratic in the number of columns of the design matrix in each step. Our algorithms support L2 regularization in the Tucker loss function, so the decompositions can readily be used in downstream learning tasks, e.g., using the factor matrix rows as embeddings for clustering [67]. Regularization also plays a critical role in the more general tensor completion problem to prevent overfitting when data is missing and has applications in differential privacy [10, 8].

The current-fastest Kronecker regression algorithm of Diao et al. [18] uses leverage score sampling and achieves the following running times for $\mathbf{A}^{(n)} \in \mathbb{R}^{I_n \times R_n}$ with $I_n \geq R_n$, for all $n \in [N]$, where $R = \prod_{n=1}^{N} R_n$ and $\omega < 2.373$ denotes the matrix multiplication exponent [4]:

1. $\tilde{O}(\sum_{n=1}^{N}(\mathrm{nnz}(\mathbf{A}^{(n)}) + R_n^\omega) + R^\omega \varepsilon^{-1})$ by sampling $\tilde{O}(R\varepsilon^{-1})$ rows of $\mathbf{K}$ by their leverage scores.

2. $\tilde{O}(\sum_{n=1}^{N}(\mathrm{nnz}(\mathbf{A}^{(n)}) + R_n^\omega \varepsilon^{-1}) + R\varepsilon^{-N})$ by sampling $\tilde{O}(R_n\varepsilon^{-1})$ rows from each factor matrix $\mathbf{A}^{(n)}$ and taking the Kronecker product of the sampled factor matrices.

Note that the second approach is linear in $R$, but the error parameter has an exponential cost in the number of factor matrices. In this work, we show that the running time of the first approach can be improved to subquadratic in $R$ without increasing the running time dependence on $\varepsilon$ in the dominant term, simultaneously improving on both approaches.

**Theorem 1.1.** *For $n \in [N]$, let $\mathbf{A}^{(n)} \in \mathbb{R}^{I_n \times R_n}$, $I_n \geq R_n$, and $\mathbf{b} \in \mathbb{R}^{I_1 \cdots I_n}$. There is a $(1 + \varepsilon)$-approximation algorithm for solving $\arg\min_{\mathbf{x}} \|(\mathbf{A}^{(1)} \otimes \cdots \otimes \mathbf{A}^{(N)})\mathbf{x} - \mathbf{b}\|_2^2$ that runs in time*

$$\tilde{O}\left(\sum_{n=1}^{N}\left(\mathrm{nnz}(\mathbf{A}^{(n)}) + R_n^\omega N^2 \varepsilon^{-2}\right) + \min_{S \subseteq [N]} \mathrm{MM}\left(\prod_{n \in S} R_n, R\varepsilon^{-1}, \prod_{n \in [N]\setminus S} R_n\right)\right), \quad (1)$$

*where $\mathrm{MM}(a, b, c)$ is the running time of multiplying an $a \times b$ matrix with a $b \times c$ matrix.*

If we do not use fast matrix multiplication (Gall and Urrutia [24] and Alman and Williams [4]), the last term in (1) is $\tilde{O}(R^2 \varepsilon^{-1})$, which is *already an improvement* over the standard $\tilde{O}(R^3 \varepsilon^{-1})$ running time. With fast matrix multiplication, $\mathrm{MM}(\prod_{n \in S} R_n, R\varepsilon^{-1}, \prod_{n \in [N]\setminus S} R_n)$ is subquadratic in $R$ for any nontrivial subset $S \notin \{\emptyset, [N]\}$, which is an improvement over $\tilde{O}(R^\omega \varepsilon^{-1}) \approx \tilde{O}(R^{2.373} \varepsilon^{-1})$. If there exists a "balanced" subset $S$ such that $\prod_{n \in S} R_n \approx \sqrt{R}$, our running time goes as low as $\tilde{O}(R^{1.626}\varepsilon^{-1})$ using [24]. For ease of notation, we denote the subquadratic improvement by the constant $\theta^* > 0$, where $R^{2-\theta^*} = \min_{S \subseteq [N]} \mathrm{MM}(\prod_{n \in S} R_n, R, \prod_{n \in [N]\setminus S} R_n)$.

Updating the core tensor in the ALS algorithm for Tucker decomposition is a pure Kronecker product regression as described in Theorem 1.1, but updating the factor matrices is a regression problem of the form $\arg\min_{\mathbf{x}} \|\mathbf{KMx} - \mathbf{b}\|_2^2$, where $\mathbf{K}$ is a Kronecker product and $\mathbf{M}$ is a matrix without any particular structure. We show that such problems can be converted to block regression problems where one of the blocks is $\mathbf{K}$. We then develop sublinear-time leverage score sampling techniques for these block matrices, which leads to the following theorem that accelerates all of the ALS steps.

**Theorem 1.2.** *There is an ALS algorithm for L2-regularized Tucker decompositions that takes a tensor $\mathcal{X} \in \mathbb{R}^{I_1 \times \cdots \times I_N}$ and returns $N$ factor matrices $\mathbf{A}^{(n)} \in \mathbb{R}^{I_n \times R_n}$ and a core tensor $\mathcal{G} \in \mathbb{R}^{R_1 \times \cdots \times R_n}$ such that each factor matrix and core update is a $(1 + \varepsilon)$-approximation to the optimum with high probability. The running times of each step are:*

- *Factor matrix $\mathbf{A}^{(k)}$: $\tilde{O}(\sum_{n=1}^{N}(\mathrm{nnz}(\mathbf{A}^{(n)}) + R_n^\omega N^2 \varepsilon^{-2}) + I_k R_{\neq k}^{2-\theta^*} \varepsilon^{-1} + I_k R \sum_{n=1}^{N} R_n + R_k^\omega)$,*

- *Core tensor $\mathcal{G}$: $\tilde{O}(\sum_{n=1}^{N}(\mathrm{nnz}(\mathbf{A}^{(n)}) + R_n^\omega N^2 \varepsilon^{-2}) + R^{2-\theta^*} \varepsilon^{-1})$,*

*where $R = \prod_{n=1}^{N} R_n$, $R_{\neq k} = R/R_k$, and $\theta^* > 0$ is a constant derived from fast rectangular matrix multiplication.*

Table 1: Running times of `TuckerALS` factor matrix and core tensor updates with different Kronecker regression methods. The factor matrices are denoted by $\mathbf{A}^{(n)} \in \mathbb{R}^{I_n \times R_n}$. The input tensor has size $I = I_1 \cdots I_N$ and the core tensor has size $R = R_1 \cdots R_N$. Let $I_{\neq k} = I/I_k$ and $R_{\neq k} = R/R_k$. We use $\omega < 2.373$ to denote the matrix-multiplication exponent and the constant $\theta^* > 0$ for the optimally balanced fast rectangular matrix multiplication as stated in Theorem 4.5, i.e., $R^{2-\theta^*} = \min_{T \subseteq [N]} \text{MM}(\prod_{n \in T} R_n, R, \prod_{n \notin T} R_n)$.

| Algorithm | Factor matrix $\mathbf{A}^{(k)}$ | Core tensor $\mathcal{G}$ |
|---|---|---|
| Naive | $O(I_k R R_{\neq k} + I_k R R_k + R_k^\omega + I R_{\neq k} + I_k R_k^2)$ | $O(R^\omega + IR)$ |
| This paper (Lemma 4.4) | $O(I_k R(\sum_{n=1}^N R_n) + R_k^\omega + I(\sum_{n \neq k} R_n) + I_k R_k^2)$ | $O(R^2 + I \sum_{n=1}^N R_n)$ |
| This paper (Theorem 1.2) | $\tilde{O}(I_k R_{\neq k}^{2-\theta^*} \varepsilon^{-1} + I_k R(\sum_{n=1}^N R_n) + R_k^\omega \varepsilon^{-2})$ | $\tilde{O}(R^{2-\theta^*} \varepsilon^{-1})$ |
| Diao et al. [18] | — | $\tilde{O}(R^\omega \varepsilon^{-2})$ |

For tensors of even modest order, the superlinear term in $R$ is the bottleneck in many applications since $R$ is exponential in the order of the tensor. It follows that our improvements are significant in both theory and practice as illustrated in our experiments in Section 6.

## 1.1 Our Contributions and Techniques

We present several new results about approximate Kronecker regression and the ALS algorithm for Tucker decompositions. Below is a summary of our contributions:

1. Our main technical contribution is the algorithm `FastKroneckerRegression` in Section 4. This Kronecker regression algorithm builds on the block-sketching tools introduced in Section 3, and combines iterative methods with a fast novel Kronecker-matrix multiplication for sparse vectors and matrices and fast rectangular matrix multiplication to achieve a running time that is *subquadratic* in the number of columns in the Kronecker matrix. A key insight is to use the original (non-sketched) Kronecker product as the preconditioner in the Richardson iterations when solving the sketched problem. This, by itself, improves the running time to quadratic. Then to achieve subqudratic running time, we exploit the singular value decomposition of Kronecker products and present a novel method for multiplying a sparsified Kronecker product matrix (Lemma 4.4 and Theorem 4.5).

2. We generalize our Kronecker regression techniques to work for Kronecker ridge regression and the factor matrix updates in ALS for Tucker decomposition. We show that a factor matrix update is equivalent to solving an *equality-constrained* Kronecker regression problem with a low-rank update to the preconditioner in the Richardson iterations. We can implement these new matrix-vector products nearly as fast by using the Woodbury matrix identity. Thus, we provably speed up each step of Tucker ALS, i.e., the core tensor and factor matrices.

3. We give a block-sketching toolkit in Section 3 that states we can sketch blocks of a matrix by their leverage scores, i.e., their leverage scores in isolation, not with respect to the entire block matrix. This is one of the ways we exploit the Kronecker product structure of the design matrix. This approach can be useful for constructing spectral approximations and for approximately solving block regression problems. One corollary is that we can use the "sketch-and-solve" method for any ridge regression problem (Corollary 3.5).

4. We compare `FastKroneckerRegression` with Diao et al. [18, Algorithm 1] on a synthetic Kronecker regression task studied in [17, 18] and as a subroutine in ALS for computing the Tucker decomposition of various image tensors [44, 50, 51]. Our results demonstrate the importance of reducing the running time dependence on the number of columns in the Kronecker product.

## 1.2 Related Work

**Kronecker Regression.** Diao et al. [17] recently gave the first Kronecker regression algorithm based on `TensorSketch` [53] that is faster than forming the Kronecker product. Diao et al. [18] improved this by removing the dependence on $O(\text{nnz}(\mathbf{b}))$ from the running time, where $\mathbf{b} \in \mathbb{R}^{I_1 \cdots I_N}$ is the response vector. Reddy, Song, and Zhang [56] recently initiated the study of *dynamic* Kronecker

regression, where the factor matrices $\mathbf{A}^{(n)}$ undergo updates and the solution vector can be efficiently queried. Marco, Martínez, and Viaña [47] studied the generalized Kronecker regression problem.

**Ridge Leverage Scores.** Alaoui and Mahoney [3] extended the notion of statistical leverage scores to account for L2 regularization. Sampling from approximate ridge leverage score distributions has since played a key role in sparse low-rank matrix approximation [16], the Nyström method [49], bounding statistical risk in ridge regression [48], and ridge regression [14, 48, 41, 33]. Fast recursive algorithms for computing approximate leverage scores [15] and for solving overconstrained least squares [40] are also closely related.

**Tensor Decomposition.** Cheng et al. [13] and Larsen and Kolda [38] used leverage score sampling to speed up ALS for CP decomposition.[2] Song et al. [59] gave a polynomial-time, relative-error approximation algorithm for several low-rank tensor decompositions, which include CP and Tucker. Frandsen and Ge [23] showed that if the tensor has an exact Tucker decomposition, then all local minima are globally optimal. Randomized low-rank Tucker decompositions based on sketching have become increasingly popular, especially in streaming applications: [45, 61, 11, 60, 31, 46, 44, 2]. The more general problem of low-rank tensor completion is also a fundamental approach for estimating the values of missing data [1, 43, 29, 28, 22]. Fundamental algorithms for tensor completion are based on ALS [68, 25, 42], Riemannian optimization [37, 34, 52], or projected gradient methods [65].

## 2 Preliminaries

**Notation.** The *order* of a tensor is the number of its dimensions. We denote scalars by normal lowercase letters $x \in \mathbb{R}$, vectors by boldface lowercase letters $\mathbf{x} \in \mathbb{R}^n$, matrices by boldface uppercase letters $\mathbf{X} \in \mathbb{R}^{m \times n}$, and higher-order tensors by boldface script letters $\mathbfcal{X} \in \mathbb{R}^{I_1 \times I_2 \times \cdots \times I_N}$. We use normal uppercase letters to denote the size of an index set (e.g., $[N] = \{1, 2, \ldots, N\}$). The $i$-th entry of a vector $\mathbf{x}$ is denoted by $x_i$, the $(i, j)$-th entry of a matrix $\mathbf{X}$ by $x_{ij}$, and the $(i, j, k)$-th entry of a third-order tensor $\mathbfcal{X}$ by $x_{ijk}$.

**Linear Algebra.** Let $\mathbf{I}_n$ denote the $n \times n$ identity matrix and $\mathbf{0}_{m \times n}$ denote the $m \times n$ zero matrix. The trans-

---

**Algorithm 1** `TuckerALS`

---

**Input:** $\mathbfcal{X} \in \mathbb{R}^{I_1 \times \cdots \times I_N}, (R_1, R_2, \ldots, R_N), \lambda$

1: Initialize core tensor $\mathbfcal{G} \in \mathbb{R}^{R_1 \times R_2 \times \cdots \times R_n}$
2: Initialize factors $\mathbf{A}^{(n)} \in \mathbb{R}^{I_n \times R_n}$ for $n \in [N]$
3: **repeat**
4:     **for** $n = 1$ to $N$ **do**
5:         $\mathbf{K} \leftarrow \mathbf{A}^{(1)} \otimes \cdots \otimes \mathbf{A}^{(n-1)} \otimes \mathbf{A}^{(n+1)} \otimes \cdots \otimes \mathbf{A}^{(N)}$
6:         $\mathbf{B} \leftarrow \mathbf{X}_{(n)}$
7:         **for** $i = 1$ to $I_n$ **do**
8:             $\mathbf{y}^* \leftarrow \arg\min_{\mathbf{y}} \|\mathbf{K}\mathbf{G}_{(n)}^{\mathsf{T}}\mathbf{y} - \mathbf{b}_{i:}^{\mathsf{T}}\|_2^2 + \lambda\|\mathbf{y}\|_2^2$
9:             Update factor row $\mathbf{a}_{i:}^{(n)} \leftarrow \mathbf{y}^{*\mathsf{T}}$
10:    $\mathbf{K} \leftarrow \mathbf{A}^{(1)} \otimes \mathbf{A}^{(2)} \otimes \cdots \otimes \mathbf{A}^{(N)}$
11:    $\mathbf{g}^* \leftarrow \arg\min_{\mathbf{g}} \|\mathbf{K}\mathbf{g} - \text{vec}(\mathbfcal{X})\|_2^2 + \lambda\|\mathbf{g}\|_2^2$
12:    Update core tensor $\mathbfcal{G} \leftarrow \text{vec}^{-1}(\mathbf{g}^*)$
13: **until** convergence
14: **return** $\mathbfcal{G}, \mathbf{A}^{(1)}, \mathbf{A}^{(2)}, \ldots, \mathbf{A}^{(N)}$

---

pose of $\mathbf{A} \in \mathbb{R}^{m \times n}$ is $\mathbf{A}^{\mathsf{T}}$, the Moore–Penrose inverse (also called pseudoinverse) is $\mathbf{A}^+$, and the spectral norm is $\|\mathbf{A}\|_2$. The singular value decomposition (SVD) of $\mathbf{A}$ is a factorization of the form $\mathbf{U}\mathbf{\Sigma}\mathbf{V}^{\mathsf{T}}$, where $\mathbf{U} \in \mathbb{R}^{m \times m}$ and $\mathbf{V} \in \mathbb{R}^{n \times n}$ are orthogonal matrices, and $\mathbf{\Sigma} \in \mathbb{R}^{m \times n}$ is a non-negative real diagonal matrix. The entries $\sigma_i(\mathbf{A})$ of $\mathbf{\Sigma}$ are the singular values of $\mathbf{A}$, and the number of non-zero singular values is equal to $r = \text{rank}(\mathbf{A})$. The *compact SVD* is a related decomposition where $\mathbf{\Sigma} \in \mathbb{R}^{r \times r}$ is a diagonal matrix containing the non-zero singular values. The Kronecker product of two matrices $\mathbf{A} \in \mathbb{R}^{m \times n}$ and $\mathbf{B} \in \mathbb{R}^{p \times q}$ is denoted by $\mathbf{A} \otimes \mathbf{B} \in \mathbb{R}^{(mp) \times (nq)}$.

**Tensor Products.** *Fibers* of a tensor are the vectors we get by fixing all but one index. If $\mathbfcal{X}$ is a third-order tensor, we denote the column, row, and tube fibers by $\mathbf{x}_{:jk}$, $\mathbf{x}_{i:k}$, and $\mathbf{x}_{ij:}$, respectively. The *mode-$n$ unfolding* of a tensor $\mathbfcal{X} \in \mathbb{R}^{I_1 \times I_2 \times \cdots \times I_N}$ is the matrix $\mathbf{X}_{(n)} \in \mathbb{R}^{I_n \times (I_1 \cdots I_{n-1} I_{n+1} \cdots I_N)}$ that arranges the mode-$n$ fibers of $\mathbfcal{X}$ as columns of $\mathbf{X}_{(n)}$ ordered lexicographically by index. The *vectorization* of $\mathbfcal{X} \in \mathbb{R}^{I_1 \times I_2 \times \cdots \times I_N}$ is the vector $\text{vec}(\mathbfcal{X}) \in \mathbb{R}^{I_1 I_2 \cdots I_N}$ formed by vertically stacking

---

[2] The design matrix in each step of ALS for CP decomposition is a Khatri–Rao product, not a Kronecker product. CP decomposition does not suffer from a bottleneck step like ALS for Tucker decomposition since it is a sparser decomposition, i.e., CP decomposition does not have a core tensor—just factor matrices.

the entries of $\mathcal{X}$ ordered lexicographically by index. For example, this transforms $\mathbf{X} \in \mathbb{R}^{m \times n}$ into a tall vector $\mathrm{vec}(\mathbf{X})$ by stacking its columns. We use $\mathrm{vec}^{-1}(\mathbf{x})$ to undo this operation when it is clear from context what the shape of the output tensor should be.

The *n-mode product* of tensor $\mathcal{X} \in \mathbb{R}^{I_1 \times I_2 \times \cdots \times I_N}$ and matrix $\mathbf{A} \in \mathbb{R}^{J \times I_n}$ is denoted by $\mathcal{Y} = \mathcal{X} \times_n \mathbf{A}$ where $\mathcal{Y} \in \mathbb{R}^{I_1 \times \cdots \times I_{n-1} \times J \times I_{n+1} \times \cdots \times I_N}$. This operation multiplies each mode-$n$ fiber of $\mathcal{X}$ by the matrix $\mathbf{A}$. This operation is expressed elementwise as

$$(\mathcal{X} \times_n \mathbf{A})_{i_1 \dots i_{n-1} j i_{n+1} \dots i_N} = \sum_{i_n=1}^{I_n} x_{i_1 i_2 \dots i_N} a_{j i_n}.$$

The Frobenius norm $\|\mathcal{X}\|_{\mathrm{F}}$ of a tensor $\mathcal{X}$ is the square root of the sum of the squares of its entries.

**Tucker Decomposition.**   The *Tucker decomposition* decomposes tensor $\mathcal{X} \in \mathbb{R}^{I_1 \times I_2 \times \cdots \times I_N}$ into a *core tensor* $\mathcal{G} \in \mathbb{R}^{R_1 \times R_2 \times \cdots \times R_N}$ and $N$ *factor matrices* $\mathbf{A}^{(n)} \in \mathbb{R}^{I_n \times R_n}$. Given a regularization parameter $\lambda \in \mathbb{R}_{\geq 0}$, we compute a Tucker decomposition by minimizing the nonconvex loss function

$$L(\mathcal{G}, \mathbf{A}^{(1)}, \dots, \mathbf{A}^{(N)}; \mathcal{X}) = \|\mathcal{X} - \mathcal{G} \times_1 \mathbf{A}^{(1)} \cdots \times_N \mathbf{A}^{(N)}\|_{\mathrm{F}}^2 + \lambda \left( \|\mathcal{G}\|_{\mathrm{F}}^2 + \sum_{n=1}^{N} \|\mathbf{A}^{(n)}\|_{\mathrm{F}}^2 \right).$$

Entries of the reconstructed tensor $\widehat{\mathcal{X}} \overset{\mathrm{def}}{=} \mathcal{G} \times_1 \mathbf{A}^{(1)} \times_2 \cdots \times_N \mathbf{A}^{(N)}$ are

$$\widehat{x}_{i_1 i_2 \dots i_N} = \sum_{r_1=1}^{R_1} \cdots \sum_{r_N=1}^{R_N} g_{r_1 r_2 \dots r_N} a_{i_1 r_1}^{(1)} \cdots a_{i_N r_N}^{(N)}. \tag{2}$$

Equation (2) demonstrates that $\widehat{\mathcal{X}}$ is the sum of $R_1 \cdots R_N$ rank-1 tensors. The tuple $(R_1, R_2, \dots, R_N)$ is the *multilinear rank* of the decomposition. The multilinear rank is typically chosen in advance and much smaller than the dimensions of $\mathcal{X}$.

**Alternating Least Squares.**   We present `TuckerALS` in Algorithm 1 and highlight its connections to Kronecker regression. The core tensor update (Lines 10–12) is a ridge regression problem where the design matrix $\mathbf{K}_{\mathrm{core}} \in \mathbb{R}^{I_1 \cdots I_N \times R_1 \cdots R_N}$ is a Kronecker product of the factor matrices. Each factor matrix update (Lines 5–9) also has Kronecker product structure, but there are additional subspace constraints we must account for. We describe these constraints in more detail in Section 5.

## 3   Row Sampling and Approximate Regression

Here we establish our sketching toolkit. The $\lambda$-*ridge leverage score* of the $i$-th row of $\mathbf{A} \in \mathbb{R}^{n \times d}$ is

$$\ell_i^\lambda(\mathbf{A}) \overset{\mathrm{def}}{=} \mathbf{a}_{i:}(\mathbf{A}^\mathsf{T}\mathbf{A} + \lambda \mathbf{I})^+ \mathbf{a}_{i:}^\mathsf{T}. \tag{3}$$

The matrix of *cross $\lambda$-ridge leverage scores* is $\mathbf{A}(\mathbf{A}^\mathsf{T}\mathbf{A} + \lambda\mathbf{I})^+\mathbf{A}^\mathsf{T}$. We denote its diagonal by $\boldsymbol{\ell}^\lambda(\mathbf{A})$ because it contains the $\lambda$-ridge leverage scores of $\mathbf{A}$. Ridge leverage scores generalize *statistical leverage scores* in that setting $\lambda = 0$ gives the leverage scores of $\mathbf{A}$. We denote the vector of statistical leverage scores by $\boldsymbol{\ell}(\mathbf{A})$. If $\mathbf{A} = \mathbf{U}\boldsymbol{\Sigma}\mathbf{V}^\mathsf{T}$ is the compact SVD of $\mathbf{A}$, then for all $i \in [n]$, we have

$$\ell_i^\lambda(\mathbf{A}) = \sum_{k=1}^{r} \frac{\sigma_k^2(\mathbf{A})}{\sigma_k^2(\mathbf{A}) + \lambda} u_{ik}^2, \tag{4}$$

where $r = \mathrm{rank}(\mathbf{A})$. It follows that every $\ell_i^\lambda(\mathbf{A}) \leq 1$ since $\mathbf{U}$ is an orthogonal matrix. We direct the reader to Alaoui and Mahoney [3] or Cohen et al. [15] for further details.

The main results in this paper build on approximate leverage score sampling for block matrices. The $\lambda$-ridge leverage scores of $\mathbf{A} \in \mathbb{R}^{n \times d}$ can be computed by appending $\sqrt{\lambda}\mathbf{I}_d$ to the bottom of $\mathbf{A}$ to get $\overline{\mathbf{A}} \in \mathbb{R}^{(n+d) \times d}$ and considering the leverage scores of $\overline{\mathbf{A}}$, so we state the following results in terms of statistical leverage scores without loss of generality.

**Definition 3.1.** For any $\mathbf{A} \in \mathbb{R}^{n \times d}$, the vector $\hat{\boldsymbol{\ell}}(\mathbf{A}) \in \mathbb{R}^n$ is a *$\beta$-overestimate* for the leverage score distribution of $\mathbf{A}$ if, for all $i \in [n]$, it satisfies

$$\frac{\hat{\ell}_i(\mathbf{A})}{\|\hat{\boldsymbol{\ell}}(\mathbf{A})\|_1} \geq \beta \frac{\ell_i(\mathbf{A})}{\|\boldsymbol{\ell}(\mathbf{A})\|_1} = \beta \frac{\ell_i(\mathbf{A})}{\mathrm{rank}(\mathbf{A})}.$$

Next we describe the approximate leverage score sampling algorithm in Woodruff [64, Section 2.4]. The core idea here is that if we sample $\tilde{O}(d/\beta)$ rows and reweight them appropriately, this smaller *sketched* matrix can be used instead of $\mathbf{A}$ to give provable guarantees for many problems.

**Definition 3.2** (Leverage score sampling). Let $\mathbf{A} \in \mathbb{R}^{n \times d}$ and $\mathbf{p} \in [0,1]^n$ be a $\beta$-overestimate for the leverage score distribution of $\mathbf{A}$ such that $\|\mathbf{p}\|_1 = 1$. $\mathtt{SampleRows}(\mathbf{A}, s, \mathbf{p})$ denotes the following procedure. Initialize sketch matrix $\mathbf{S} = \mathbf{0}_{s \times n}$. For each row $i$ of $\mathbf{S}$, independently and with replacement, select an index $j \in [n]$ with probability $p_j$ and set $s_{ij} = 1/\sqrt{p_j s}$. Return sketch $\mathbf{S}$.

The main result in this section is that we can choose to sketch a single block of a matrix by the leverage scores of that block in isolation. This sketched submatrix can then be used with the other (non-sketched) block to give a spectral approximation to the original matrix or for approximate linear regression. The notation $\mathbf{A} \preccurlyeq \mathbf{B}$ is the Loewner order and means $\mathbf{B} - \mathbf{A}$ is positive semidefinite.

**Lemma 3.3.** *Let* $\mathbf{A} = [\mathbf{A}_1; \mathbf{A}_2]$ *be vertically stacked with* $\mathbf{A}_1 \in \mathbb{R}^{n_1 \times d}$ *and* $\mathbf{A}_2 \in \mathbb{R}^{n_2 \times d}$. *Let* $\mathbf{p} \in [0,1]^{n_1}$ *be a $\beta$-overestimate for the leverage score distribution of* $\mathbf{A}_1$. *If* $s > 144d\ln(2d/\delta)/(\beta\varepsilon^2)$, *the sketch* $\mathbf{S}$ *returned by* $\mathtt{SampleRows}(\mathbf{A}_1, s, \mathbf{p})$ *guarantees, with probability at least* $1 - \delta$, *that*

$$(1 - \varepsilon)\mathbf{A}^\mathsf{T}\mathbf{A} \preccurlyeq (\mathbf{S}\mathbf{A}_1)^\mathsf{T}\mathbf{S}\mathbf{A}_1 + \mathbf{A}_2^\mathsf{T}\mathbf{A}_2 \preccurlyeq (1 + \varepsilon)\mathbf{A}^\mathsf{T}\mathbf{A}.$$

**Lemma 3.4** (Approximate block regression). *Consider the problem* $\arg\min_{\mathbf{x} \in \mathbb{R}^d} \|\mathbf{A}\mathbf{x} - \mathbf{b}\|_2^2$ *where* $\mathbf{A} = [\mathbf{A}_1; \mathbf{A}_2]$ *and* $\mathbf{b} = [\mathbf{b}_1; \mathbf{b}_2]$ *are vertically stacked and* $\mathbf{A}_1 \in \mathbb{R}^{n_1 \times d}$, $\mathbf{A}_2 \in \mathbb{R}^{n_2 \times d}$, $\mathbf{b}_1 \in \mathbb{R}^{n_1}$, $\mathbf{b}_2 \in \mathbb{R}^{n_2}$. *Let* $\mathbf{p} \in [0,1]^{n_1}$ *be a $\beta$-overestimate for the leverage score distribution of* $\mathbf{A}_1$. *Let* $s \geq 1680d\ln(40d)/(\beta\varepsilon)$ *and let* $\mathbf{S}$ *be the output of* $\mathtt{SampleRows}(\mathbf{A}_1, s, \mathbf{p})$. *If*

$$\tilde{\mathbf{x}}^* = \arg\min_{\mathbf{x} \in \mathbb{R}^d}\Big(\|\mathbf{S}(\mathbf{A}_1\mathbf{x} - \mathbf{b}_1)\|_2^2 + \|\mathbf{A}_2\mathbf{x} - \mathbf{b}_2\|_2^2\Big),$$

*then, with probability at least* $9/10$, *we have*

$$\|\mathbf{A}\tilde{\mathbf{x}}^* - \mathbf{b}\|_2^2 \leq (1 + \varepsilon)\min_{\mathbf{x} \in \mathbb{R}^d}\|\mathbf{A}\mathbf{x} - \mathbf{b}\|_2^2.$$

We defer the proofs of these results to Appendix A. The key idea behind Lemma 3.4 is that leverage scores do not increase if rows are appended to the matrix. This then allows us to prove a sketched submatrix version of Drineas et al. [19, Lemma 8] for approximate matrix multiplication and satisfy the structural conditions for approximate least squares in Drineas et al. [20]. One consequence is that we can "sketch and solve" ridge regression, which was shown in [63, Theorem 1] and [6, Theorem 2].

**Corollary 3.5.** *For any* $\mathbf{A} \in \mathbb{R}^{n \times d}$, $\mathbf{b} \in \mathbb{R}^d$, $\lambda \geq 0$, *consider*

$$\arg\min_{\mathbf{x} \in \mathbb{R}^d}(\|\mathbf{A}\mathbf{x} - \mathbf{b}\|_2^2 + \lambda\|\mathbf{x}\|_2^2).$$

*Let* $\mathbf{p} \in [0,1]^{n_1}$ *be a $\beta$-overestimate for the leverage scores of* $\mathbf{A}$ *and* $s \geq 1680d\ln(40d)/(\beta\varepsilon)$. *If* $\mathbf{S}$ *is the output of* $\mathtt{SampleRows}(\mathbf{A}, s, \mathbf{p})$, *then, with probability at least* $9/10$, *the sketched solution*

$$\tilde{\mathbf{x}}^* = \arg\min_{\mathbf{x} \in \mathbb{R}^d}(\|\mathbf{S}(\mathbf{A}\mathbf{x} - \mathbf{b})\|_2^2 + \lambda\|\mathbf{x}\|_2^2)$$

*gives a $(1 + \varepsilon)$-approximation to the original problem.*

**Remark 3.6.** The success probability of the sketch can be boosted from $9/10$ to $1 - \delta$ by sampling a factor of $O(\log(1/\delta))$ more rows. See the discussion in Chen and Price [12, Section 2] about matrix concentration bounds for more details.

## 4 Kronecker Regression

Now we describe the key ingredients that allow us to design an approximate Kronecker regression algorithm whose running time is *subquadratic* in the number of columns in the design matrix.

1. The leverage score distribution of a Kronecker product matrix $\mathbf{K} = \mathbf{A}^{(1)} \otimes \cdots \otimes \mathbf{A}^{(N)}$ is a *product distribution* of the leverage score distributions of its factor matrices. Therefore, we can sample rows of $\mathbf{K}$ from $\boldsymbol{\ell}(\mathbf{K})$ with replacement in $\tilde{O}(N)$ time after a preprocessing step.

2. The normal matrix $\mathbf{K}^\mathsf{T}\mathbf{K} + \lambda\mathbf{I}$ in the ridge regression problem $\min_{\mathbf{x}} \|\mathbf{K}\mathbf{x} - \mathbf{b}\|_2^2 + \lambda\|\mathbf{x}\|_2^2$ is a $O(1)$-spectral approximation of the sketched matrix $(\mathbf{S}\mathbf{K})^\mathsf{T}\mathbf{S}\mathbf{K} + \lambda\mathbf{I}$ by Lemma 3.3. Thus we can use Richardson iteration with $(\mathbf{K}^\mathsf{T}\mathbf{K} + \lambda\mathbf{I})^+$ as the preconditioner to *solve the sketched instance*, which guarantees a $(1 + \varepsilon)$-approximation. Using $(\mathbf{K}^\mathsf{T}\mathbf{K} + \lambda\mathbf{I})^+$ as the preconditioner allows us to *heavily exploit the Kronecker structure* with fast matrix-vector multiplications.

3. At this point, *Kronecker matrix-vector multiplications* are still the bottleneck, so we partition the factor matrices into two groups by their number of columns and use our novel way of multiplying sparsified Kronecker product matrices as well as fast rectangular matrix multiplication to get a subquadratic running time.

This first result shows how $\lambda$-ridge leverage scores of a Kronecker product matrix decompose according to the SVDs of its factor matrices. All missing proofs in this section are deferred to Appendix B.

**Lemma 4.1.** *Let* $\mathbf{K} = \mathbf{A}^{(1)} \otimes \mathbf{A}^{(2)} \otimes \cdots \otimes \mathbf{A}^{(N)}$, *where each factor matrix* $\mathbf{A}^{(n)} \in \mathbb{R}^{I_n \times R_n}$. *Let* $(i_1, i_2, \ldots, i_N)$ *be the natural row indexing of* $\mathbf{K}$ *by its factors. Let the factor SVDs be* $\mathbf{A}^{(n)} = \mathbf{U}^{(n)}\mathbf{\Sigma}^{(n)}\mathbf{V}^{(n)\mathsf{T}}$. *For any* $\lambda \geq 0$, *the* $\lambda$-*ridge leverage scores of* $\mathbf{K}$ *are*

$$\ell_{(i_1,\ldots,i_N)}^\lambda(\mathbf{K}) = \sum_{\mathbf{t} \in T} \frac{\prod_{n=1}^N \sigma_{t_n}^2(\mathbf{A}^{(n)})}{\prod_{n=1}^N \sigma_{t_n}^2(\mathbf{A}^{(n)}) + \lambda} \left( \prod_{n=1}^N u_{i_n t_n}^{(n)} \right)^2, \tag{5}$$

*where the sum is over* $T = [R_1] \times [R_2] \times \cdots \times [R_N]$. *For statistical leverage scores, this simplifies to* $\ell_{(i_1,\ldots,i_N)}(\mathbf{K}) = \prod_{n=1}^N \ell_{i_n}(\mathbf{A}^{(n)})$.

This proof repeatedly uses the mixed-product property for Kronecker products and the definition of $\lambda$-ridge leverage scores in Equation (3).

## 4.1 Iterative Methods

Now we state a result for the convergence rate of preconditioned Richardson iteration [57], which uses the notation $\|\mathbf{x}\|_\mathbf{M}^2 = \mathbf{x}^\mathsf{T}\mathbf{M}\mathbf{x}$.

**Lemma 4.2** (Preconditioned Richardson iteration). *Let* $\mathbf{M}$ *be any matrix such that* $\mathbf{A}^\mathsf{T}\mathbf{A} \preccurlyeq \mathbf{M} \preccurlyeq \kappa \cdot \mathbf{A}^\mathsf{T}\mathbf{A}$ *for some* $\kappa \geq 1$. *Let* $\mathbf{x}^{(k+1)} = \mathbf{x}^{(k)} - \mathbf{M}^+(\mathbf{A}^\mathsf{T}\mathbf{A}\mathbf{x}^{(k)} - \mathbf{A}^\mathsf{T}\mathbf{b})$. *Then,*

$$\|\mathbf{x}^{(k)} - \mathbf{x}^*\|_\mathbf{M} \leq (1 - 1/\kappa)^k \|\mathbf{x}^{(0)} - \mathbf{x}^*\|_\mathbf{M},$$

*where* $\mathbf{x}^* = \arg\min_{\mathbf{x} \in \mathbb{R}^d} \|\mathbf{A}\mathbf{x} - \mathbf{b}\|_2^2$.

**Remark 4.3.** The ridge regression algorithm in Chowdhury et al. [14] is also based on sketching and preconditioned Richardson iteration. They consider short and wide matrices where $d \gg n$ and use the *sketched normal matrix as the preconditioner* to solve the original problem. One of our main technical contributions is to use the *original normal matrix as the preconditioner* to solve the sketched problem. Reversing this is advantageous because computing the psueduoinverse and matrix-vector products with the original Kronecker matrix is substantially less expensive due to its Kronecker structure. However, this still motivates the need for faster Kronecker matrix-vector multiplications.

## 4.2 Fast Kronecker-Matrix Multiplication

The next result is a simple but useful observation about extracting the rightmost factor matrix from the Kronecker product and recursively computing a new less expensive Kronecker-matrix multiplication.

**Lemma 4.4.** *Let* $\mathbf{A}^{(n)} \in \mathbb{R}^{I_n \times J_n}$, *for* $n \in [N]$, *and* $\mathbf{B} \in \mathbb{R}^{J_1 \cdots J_N \times K}$. *There is an algorithm* `KronMatMul`$([\mathbf{A}^{(1)}, \ldots, \mathbf{A}^{(N)}], \mathbf{B})$ *that computes* $(\mathbf{A}^{(1)} \otimes \mathbf{A}^{(2)} \otimes \cdots \otimes \mathbf{A}^{(N)})\mathbf{B} \in \mathbb{R}^{(I_1 \cdots I_N) \times K}$ *in* $O(K \sum_{n=1}^N J_1 \cdots J_n I_n \cdots I_N)$ *time.*

The following theorem is more sophisticated. We write the statement in terms of rectangular matrix multiplication time $\text{MM}(a, b, c)$, which is the time to multiply an $a \times b$ matrix by a $b \times c$ matrix.

**Theorem 4.5.** *Let* $\mathbf{A}^{(n)} \in \mathbb{R}^{I_n \times R_n}$, *for* $n \in [N]$, $I = I_1 \cdots I_N$, $R = R_1 \cdots R_N$, $\mathbf{b} \in \mathbb{R}^I$, $\mathbf{c} \in \mathbb{R}^R$, *and* $\mathbf{S} \in \mathbb{R}^{I \times I}$ *be a diagonal matrix with* $\tilde{O}(R\varepsilon^{-1})$ *nonzeros. The vectors*

$$(\mathbf{A}_1 \otimes \cdots \otimes \mathbf{A}_N)^\mathsf{T}\mathbf{S}\mathbf{b} \quad and \quad \mathbf{S}(\mathbf{A}_1 \otimes \cdots \otimes \mathbf{A}_N)\mathbf{c}$$

*can be computed in time* $\tilde{O}(\min_{T \subseteq [N]} \text{MM}(\prod_{n \in T} R_n, R\varepsilon^{-1}, \prod_{n \notin T} R_n))$.

---

**Algorithm 2** `FastKroneckerRegression`

---

**Input:** Factor matrices $\mathbf{A}^{(n)} \in \mathbb{R}^{I_n \times R_n}$, response vector $\mathbf{b} \in \mathbb{R}^{I_1 \cdots I_N}$, L2 regularization strength $\lambda$, error $\varepsilon$, failure probability $\delta$

1: Set $R \leftarrow R_1 R_2 \cdots R_N$
2: **for** $n = 1$ to $N$ **do**
3:     Compute a spectral approximation $\tilde{\mathbf{A}}^{(n)}$ with $\tilde{O}(R_n N^2 \varepsilon^{-2})$ rows by Lemma 3.3 such that

$$\mathbf{A}^{(n)\mathsf{T}} \mathbf{A}^{(n)} \preccurlyeq \tilde{\mathbf{A}}^{(n)\mathsf{T}} \tilde{\mathbf{A}}^{(n)} \preccurlyeq (1 + \log(1 + \varepsilon/4)/N) \mathbf{A}^{(n)\mathsf{T}} \mathbf{A}^{(n)} \tag{6}$$

4:     Compute $\tilde{\mathbf{A}}^{(n)\mathsf{T}} \tilde{\mathbf{A}}^{(n)}$ and the SVD of $\tilde{\mathbf{A}}^{(n)\mathsf{T}} \tilde{\mathbf{A}}^{(n)} = \mathbf{V}^{(n)} (\mathbf{\Sigma}^{(n)\mathsf{T}} \mathbf{\Sigma}^{(n)}) \mathbf{V}^{(n)\mathsf{T}}$
5:     Compute $(1 + \log(1 + \varepsilon/2)/N)$-approximate leverage scores $\boldsymbol{\ell}(\mathbf{A}^{(n)})$ using Lemma B.4 by applying a random Johnson–Lindenstrauss projection
6: Initialize product distribution data structure $\mathcal{P}$ to sample indices from $(\boldsymbol{\ell}(\mathbf{A}^{(1)}), \cdots, \boldsymbol{\ell}(\mathbf{A}^{(N)}))$
7: Set $\mathbf{D} \leftarrow (\mathbf{\Sigma}^{(1)\mathsf{T}} \mathbf{\Sigma}^{(1)} \otimes \cdots \otimes \mathbf{\Sigma}^{(N)\mathsf{T}} \mathbf{\Sigma}^{(N)} + \lambda \mathbf{I}_R)^+$
8: Let $\mathbf{M}^+ = (\mathbf{V}^{(1)} \otimes \cdots \otimes \mathbf{V}^{(N)}) \mathbf{D} (\mathbf{V}^{(1)} \otimes \cdots \otimes \mathbf{V}^{(N)})^{\mathsf{T}}$
9: Set $s \leftarrow \lceil 1680 R \ln(40R) \ln(1/\delta)/\varepsilon \rceil$
10: Set $\mathbf{S} \leftarrow$ `SampleRows`$(\mathbf{K}, s, \mathcal{P})$
11: Let $\tilde{\mathbf{K}} = \mathbf{SK}$ and $\tilde{\mathbf{b}} = \mathbf{Sb}$
12: Initialize $\mathbf{x} \leftarrow \mathbf{0}_R$
13: **repeat**
14:     $\mathbf{x} \leftarrow \mathbf{x} - (1 - \sqrt{\varepsilon}) \mathbf{M}^+ (\tilde{\mathbf{K}}^{\mathsf{T}} \tilde{\mathbf{K}} \mathbf{x} + \lambda \mathbf{x} - \tilde{\mathbf{K}}^{\mathsf{T}} \tilde{\mathbf{b}})$ using fast Kronecker-matrix multiplication
15: **until** convergence
16: **return** $\mathbf{x}$

---

The core idea behind Theorem 4.5 is that the factor matrices can be partitioned into two groups to achieve a good "column-product" balance, i.e., $\min_{T \subseteq [N]} \max\{\prod_{n \in T} R_n, \prod_{n \notin T} R_n\}$ is close to $\sqrt{R}$. Then we use the fact that $\text{nnz}(\mathbf{S}) = \tilde{O}(R \varepsilon^{-1})$ with a sparsity-aware `KronMatMul` to solve each part of this partition separately, and combine them with fast rectangular matrix multiplication. If we achieve perfect balance, the running time is $\tilde{O}(R^{1.626} \varepsilon^{-1})$ using results of Gall and Urrutia [24], which are explained in detail in van den Brand and Nanongkai [62, Appendix C]. If one of these two factor matrix groups has at most 0.9 of the "column-product mass," the running time is $\tilde{O}(R^{1.9} \varepsilon^{-1})$.

### 4.3  Main Algorithm

We are now ready to present our main algorithm for solving approximate Kronecker regression.

**Theorem 4.6.** *For any Kronecker product matrix* $\mathbf{K} = \mathbf{A}^{(1)} \otimes \cdots \otimes \mathbf{A}^{(N)} \in \mathbb{R}^{I_1 \cdots I_N \times R_1 \cdots R_N}$, $\mathbf{b} \in \mathbb{R}^{I_1 \cdots I_N}$, $\lambda \geq 0$, $\varepsilon \in (0, 1/4]$, *and* $\delta > 0$, `FastKroneckerRegression` *returns* $\mathbf{x}^* \in \mathbb{R}^{R_1 \cdots R_N}$ *in*

$$\tilde{O}\left( \sum_{n=1}^{N} \big( \text{nnz}(\mathbf{A}^{(n)}) + R_n^\omega N^2 \varepsilon^{-2} \big) + \min_{S \subseteq [N]} \text{MM}\Big( \prod_{n \in S} R_n, R\varepsilon^{-1}, \prod_{n \in [N] \setminus S} R_n \Big) \right),$$

*time such that, with probability at least* $1 - \delta$,

$$\|\mathbf{Kx}^* - \mathbf{b}\|_2^2 + \lambda \|\mathbf{x}\|_2^2 \leq (1 + \varepsilon) \min_{\mathbf{x}} \|\mathbf{Kx} - \mathbf{b}\|_2^2 + \lambda \|\mathbf{x}\|_2^2.$$

We defer the proof to Appendix B.2 and sketch how the ideas in Algorithm 2 come together. First, we do not compute the pseudoinverse $\tilde{\mathbf{K}}^+$ but instead use iterative Richardson iteration (Lemma 4.2), which allows us avoid a $\tilde{O}(R^\omega \varepsilon^{-1})$ running time. This technique by itself, however, only allows us to reduce the running time to $\tilde{O}(R^2 \varepsilon^{-1})$ since all of the matrix-vector products (e.g., $\tilde{\mathbf{K}}^{\mathsf{T}} \tilde{\mathbf{b}}$, $\tilde{\mathbf{K}} \mathbf{x}$, and multiplication against $\mathbf{M}^+$) naively take $\Omega(R^2)$ time. To achieve subquadratic time, we need three more ideas: (1) compute an approximate SVD of each Gram matrix $\mathbf{A}^{(n)\mathsf{T}} \mathbf{A}^{(n)}$ in order to construct the decomposed preconditioner $\mathbf{M}^+$; (2) use fast Kronecker-vector multiplication (e.g., Lemma 4.4) to exploit the Kronecker structure of the decomposed preconditioner; (3) noting that Lemma 4.4 for the Kronecker-vector products $\tilde{\mathbf{K}}^{\mathsf{T}} \tilde{\mathbf{b}}$ and $\tilde{\mathbf{K}}^{\mathsf{T}}(\tilde{\mathbf{K}} \mathbf{x})$ is insufficient because the intermediate vectors can be large, we develop a novel multiplication algorithm in Theorem 4.5 that fully exploits the sparsity, Kronecker structure, and fast rectangular matrix multiplication of Gall and Urrutia [24].

# 5 Applications to Low-Rank Tucker Decomposition

Now we apply our fast Kronecker regression algorithm to `TuckerALS` and prove Theorem 1.2. We list the running times of different factor matrix and core update algorithms in Table 1, and we analyze these subroutines in Appendix C.3.

**Core Tensor Update.**    The core update running time in Theorem 1.2 is a direct consequence of our algorithm for fast Kronecker regression in Theorem 4.6. The only difference is that we avoid recomputing the SVD and Gram matrix of each factor since these are computed at the end of each factor matrix update and stored for future use.

**Factor Matrix Update.**    The factor matrix updates require more work because of the $\mathbf{G}_{(n)}^{\mathsf{T}}\mathbf{y}$ term in Line 8 of `TuckerALS`. To overcome this, we substitute variables and recast each factor update as an equality-constrained Kronecker regression problem with an appended low-rank block to account for the L2 regularization of the original variables. To support this new low-rank block, we use the *Woodbury matrix identity* to extend the technique of using Richardson iterations with fast Kronecker matrix-vector multiplication for solving sketched regression instances.

The next result formalizes this substitution and reduces the problem to block Kronecker regression with a subspace constraint. This result relies on the fact that the least squares solution to $\|\mathbf{M}\mathbf{x} - \mathbf{z}\|_2^2$ with minimum norm is $\mathbf{M}^+\mathbf{z}$.

**Lemma 5.1.** *Let* $\mathbf{A} \in \mathbb{R}^{n \times m}$, $\mathbf{M} \in \mathbb{R}^{m \times d}$, $\mathbf{b} \in \mathbb{R}^n$, *and* $\lambda \geq 0$. *For any ridge regression problem of the form* $\arg\min_{\mathbf{x} \in \mathbb{R}^d}(\|\mathbf{A}\mathbf{M}\mathbf{x} - \mathbf{b}\|_2^2 + \lambda\|\mathbf{x}\|_2^2)$, *we can solve*

$$\mathbf{z}_{\text{opt}} = \arg\min_{\mathbf{N}\mathbf{z}=\mathbf{0}}\|\mathbf{A}\mathbf{z} - \mathbf{b}\|_2^2 + \lambda\|\mathbf{M}^+\mathbf{z}\|_2^2,$$

*where* $\mathbf{N} = \mathbf{I}_m - \mathbf{M}\mathbf{M}^+$, *and return vector* $\mathbf{M}^+\mathbf{z}_{\text{opt}}$ *instead.*

*Proof.* Let $\mathbf{z} = \mathbf{M}\mathbf{x} \in \mathbb{R}^m$. For any $\mathbf{x} \in \mathbb{R}^d$, $\mathbf{z}$ is in the column space of $\mathbf{M}$ and hence orthogonal to any vector in the left null space of $\mathbf{M}$. Therefore, we can optimize over $\mathbf{z} \in \mathbb{R}^m$ subject to $\mathbf{N}\mathbf{z} = \mathbf{0}$ instead because for any $\mathbf{x} \in \mathbb{R}^d$, $\mathbf{N}\mathbf{M}\mathbf{x} = (\mathbf{I}_m - \mathbf{M}\mathbf{M}^+)\mathbf{M}\mathbf{x} = (\mathbf{M} - \mathbf{M})\mathbf{x} = \mathbf{0}$. Using this substitution, we can also replace the term $\lambda\|\mathbf{x}\|_2^2$ by $\lambda\|\mathbf{M}^+\mathbf{z}\|_2^2$ because for any $\mathbf{z}$, the least squares solution to $\mathbf{z} = \mathbf{M}\mathbf{x}$ with minimum norm is $\mathbf{M}^+\mathbf{z}$ [54]. $\square$

To solve this constrained regression problem, we can add a scaled version of the constraint matrix $\mathbf{N}$ as a block to the approximate regression problem and take the projection of the resulting solution.

**Lemma 5.2** (Approximate equality-constrained regression). *Let* $\mathbf{M} \in \mathbb{R}^{n \times d}$, $\mathbf{N} \in \mathbb{R}^{m \times d}$, $\mathbf{b} \in \mathbb{R}^n$, *and* $0 < \varepsilon < 1/3$. *To solve* $\min_{\mathbf{N}\mathbf{x}=\mathbf{0}}\|\mathbf{M}\mathbf{x} - \mathbf{b}\|_2^2$ *to a* $(1 + \varepsilon)$-*approximation, it suffices to solve*

$$\min_{\mathbf{x} \in \mathbb{R}^d}\left\|\begin{bmatrix}\mathbf{M}\\\sqrt{w}\mathbf{N}\end{bmatrix}\mathbf{x} - \begin{bmatrix}\mathbf{b}\\\mathbf{0}\end{bmatrix}\right\|_2^2$$

*to a* $(1 + \varepsilon/3)$-*approximation with* $w \geq (1 + 12/\varepsilon)\|\mathbf{M}\mathbf{N}^+\|_2^2$.

Letting $\mathbf{z} = \mathbf{G}_{(n)}^{\mathsf{T}}\mathbf{y}$ in Line 8 of `TuckerALS` and modifying `FastKroneckerRegression` to support additional low-rank updates to the preconditioner, we get the `FastFactorMatrixUpdate` algorithm, presented as Algorithm 3 in Appendix C.2. The analysis is similar to the proofs of Theorem 4.6. The factor matrix updates benefit in the same way as before from fast Kronecker matrix-vector products, and new low-rank block updates are supported via the Woodbury identity. We defer the proofs of the next two results to Appendix C.

**Theorem 5.3.** *For any* $\lambda \geq 0$, $\varepsilon \in (0, 1/3)$, *and* $\delta > 0$, *the* `FastFactorMatrixUpdate` *algorithm updates* $\mathbf{A}_{(k)} \in \mathbb{R}^{I_k \times R_k}$ *in* `TuckerALS` *with a* $(1 + \varepsilon)$-*approximation, with probability at least* $1 - \delta$, *in time*

$$\tilde{O}\left(I_k R_{\neq k}^2 \varepsilon^{-1}\log(1/\delta) + I_k R\sum_{n=1}^N R_n + R_k^\omega \varepsilon^{-2}\right).$$

**Corollary 5.4.** `FastFactorMatrixUpdate` *updates* $\mathbf{A}^{(k)} \in \mathbb{R}^{I_k \times R_k}$ *in* $\tilde{O}(I_k R_{\neq k}^{2-\theta^*}\varepsilon^{-1}\log(1/\delta) + I_k R\sum_{n=1}^N R_n + R_k^\omega\varepsilon^{-2})$ *time, where* $\theta^* > 0$ *is the optimally balanced* MM *exponent in Theorem 4.5.*

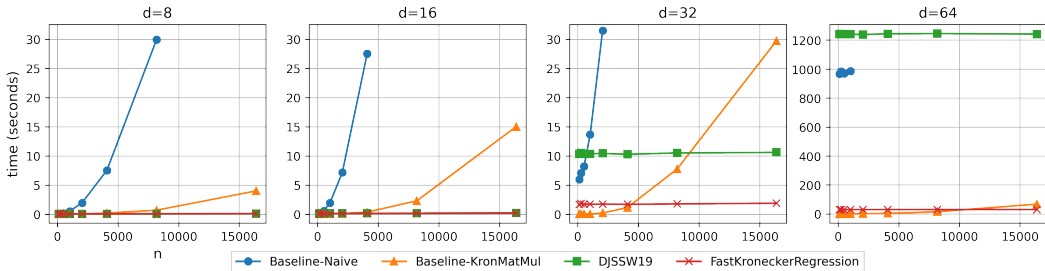

Figure 1: Running times of Kronecker regression algorithms with a design matrix of size $n^2 \times d^2$.

## 6 Experiments

All experiments were run using NumPy [26] with an Intel Xeon W-2135 processor (8.25MB cache, 3.70 GHz) and 128GB of RAM. The `FastKroneckerRegression`-based ALS experiments for low-rank Tucker decomposition on image tensors are deferred to Appendix D.2. All of our code is available at `https://github.com/fahrbach/subquadratic-kronecker-regression`.

**Kronecker regression.** We build on the numerical experiments in [17, 18] for Kronecker regression that use two random factor matrices. We generate matrices $\mathbf{A}^{(1)}, \mathbf{A}^{(2)} \in \mathbb{R}^{n \times d}$ where each entry is drawn i.i.d. from the normal distribution $\mathcal{N}(1, 0.001)$ and compare several algorithms for solving $\min_{\mathbf{x}} \|(\mathbf{A}^{(1)} \otimes \mathbf{A}^{(2)})\mathbf{x} - \mathbf{1}_{n^2}\|_2^2 + \lambda \|\mathbf{x}\|_2^2$ as we increase $n, d$. The running times are plotted in Figure 1.

The algorithms we compare are: (1) a baseline that solves the normal equation $(\mathbf{K}^\intercal \mathbf{K} + \lambda \mathbf{I})^+ \mathbf{K}^\intercal \mathbf{b}$ and fully exploits the Kronecker structure of $\mathbf{K}^\intercal \mathbf{K}$ before calling `np.linalg.pinv()`; (2) an enhanced baseline that combines the SVDs of $\mathbf{A}^{(n)}$ with Lemma 4.4, e.g., `KronMatMul([(U^(1))^⊺, (U^(2))^⊺], b)`, using only Kronecker-vector products; (3) the sketching algorithm of Diao et al. [18, Algorithm 1]; and (4) our `FastKroneckerRegression` algorithm in Algorithm 2. For both sketching algorithms, we use $\varepsilon = 0.1$ and $\delta = 0.01$. We reduce the number of row samples in both algorithms by $\alpha = 10^{-5}$ so that the algorithms are more practical and comparable to the earlier experiments in [17, 18]. Lastly, we set $\lambda = 10^{-3}$. We discuss additional parameter choice details and the full results in Appendix D.1.

The running times in Figure 1 demonstrate several different behaviors. The naive baseline quickly becomes impractical for moderately large values of $n$ or $d$. `KronMatMul` is competitive for $n \leq 10^4$, especially since it is an exact method. The runtimes of the sketching algorithms are nearly-independent of $n$. Diao et al. [18] works well for small $d$, but deteriorates tremendously as $d$ grows because it computes $((\mathbf{SK})^\intercal \mathbf{SK} + \lambda \mathbf{I})^+ \in \mathbb{R}^{d^2 \times d^2}$ and cannot exploit the Kronecker structure of $\mathbf{K}$, which takes $O(d^6)$ time. `FastKroneckerRegression`, on the other hand, runs in $O(d^4)$ time because it uses quadratic-time Kronecker-vector products in each Richardson iteration step (Line 14).

Table 2: Kronecker regression losses for $d = 64$. OPT denotes the loss of the `KronMatMul` algorithm, `DJSSW19` is Diao et al. [18, Algorithm 1], and Algorithm 2 is `FastKroneckerRegression`. We also record the relative error of each algorithm and the number of rows sampled from $\mathbf{A}^{(1)} \otimes \mathbf{A}^{(2)}$.

| $n$ | OPT | Algorithm 2 | Approx | DJSSW19 | Approx | Rows sampled (%) |
|---|---|---|---|---|---|---|
| 1024 | 0.031 | 0.032 | 1.051 | 0.035 | 1.138 | 0.0370 |
| 2048 | 0.123 | 0.126 | 1.026 | 1.577 | 12.792 | 0.0093 |
| 4096 | 0.507 | 0.520 | 1.026 | 275.566 | 543.776 | 0.0023 |
| 8192 | 2.073 | 2.136 | 1.030 | 333.430 | 160.809 | 0.0006 |
| 16384 | 8.238 | 8.608 | 1.045 | 546391.728 | 66329.791 | 0.0001 |

These experiments also show that combining sketching with iterative methods can give better *sketch efficiency*. Table 2 compares the loss of [18, Algorithm 1] and `FastKroneckerRegression` to an exact baseline OPT for $d = 64$. Both algorithms use the exact same sketch $\mathbf{SK}$ for each value of $n$. Our algorithm uses the original $(\mathbf{K}^\intercal \mathbf{K} + \lambda \mathbf{I})^+$ as a preconditioner to solve the sketched problem, whereas Diao et al. [18, Algorithm 1] computes $((\mathbf{SK})^\intercal \mathbf{SK} + \lambda \mathbf{I})^+ (\mathbf{SK})^\intercal \mathbf{Sb}$ exactly and becomes numerically unstable for $n \geq 2048$ when $d \in \{16, 32, 64\}$. This raises the question about how to combine sketched information with the original data to achieve more efficient algorithms, even when solving sketched instances. We leave this question of sketch efficiency as an interesting future work.

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
