# OpenReview forum: "Subquadratic Kronecker Regression with Applications to Tensor Decomposition"
_NeurIPS.cc/2022/Conference — NeurIPS 2022 Accept_

### Official Review · Reviewer_yEQL · 2022-07-07

**Rating:** 7
**Confidence:** 4
**Soundness:** 3 good
**Presentation:** 3 good
**Contribution:** 3 good

**Summary:**

In this paper, the authors study fast sketching based algorithms for Kronecker Product Regression and show how this algorithm can be used to speedup Alternating Least Squares (ALS) algorithm for computing Tucker Decomposition. The main improvement over previous work of Diao et. al. [16], which introduced leverage score sampling for Kronecker Product Regression seems to be recognizing that the matrix $M = (K^{\intercal}K +\lambda  I$) can be used as a preconditioner for solving the sketched ridge regression using Richardson's Iteration and showing a way to quickly compute $(K^{\intercal}K +\lambda  I)^+v$ for a vector $v$ using fast matrix multiplication by appropriately dividing the matrices in the Kronecker Product into two disjoint subsets.. Although updating the core tensor in Tucker Decomposition requires one to solve a simple Kronecker Product Ridge Regression problem, the factor matrix update requires one to solve a more complicated Kronecker Product Ridge Regression with a subspace constraint. The authors show that their techniques can be used to solve this version of the Kronecker Product Ridge Regression and therefore all the steps in ALS algorithm for computing Tucker Decomposition are made faster.

**Questions:**

- Can you clarify my doubts about the input sparsity claims in Theorem 1.1?
- I am confused about where Theorem 4.5 is used in $\texttt{FastKroneckerRegression}$. Isn't the only product of a vector with a kronecker product in line 13? And the vector there isn't sparse. Am I missing anything in regards to where it is used?
- If I understand correctly, the main speedup over [16] is obtained by not computing the product $\tilde{K}^{\intercal} \times \tilde{K}$ and instead using the Richardson's iteration which using Fast Kronecker Product you can perform quickly? Can you explain how the term $\tilde{O}(\text{MM}(\Pi_{n \in T} R_n, R\varepsilon^{-1}, \Pi_{n \notin T}R_n))$ arises in line 13?
- How does solving intermediate problems of ALS to a small approximation affect the overall convergence? I'd think it shouldn't be affected too much but do you have any small experiments where you observe anything significant?
- Please let me know if I am misunderstanding or missing anything. I can take a closer look at sections in the appendix if they are explained there better.

**Strengths And Weaknesses:**

Strengths:
- Shows how speeding up Kronecker Product Ridge regression helps in running ALS for Tucker Decomposition. This wasn't observed in the previous works [15,16].
- Improving the running times over previous work by observing that solving the sketched problem using Richardson Iteration with $(K^{\intercal}K + \lambda I)^{+}$ is preconditioner leads to faster running times.

Weaknesses:
- The paper doesn't have any new contributions for sketching literature. Note that sketching a submatrix $A_1$ by leverage score sampling rows of $A_1$ using leverage scores with respect to $A_1$ obviously works because of the property that leverage scores only decrease when the new rows are added. This observation is not new and can be seen in the paper "Sharper Bounds for Regularized Data Fitting".
- It is not clear how the authors claim input sparsity running times for Kronecker Product Ridge Regression when in their algorithm, they compute singular value decompositions of $A_1,\ldots, A_N$. If one doesn't compute singular value decompositions, how is $(K^{\intercal}K + \lambda I)^{+}$ used as a preconditioner? Am I missing something here? I understand input sparsity time algorithms are probably not necessary for intermediate problems of ALS for tucker decomposition as one cannot guarantee that the intermediate matrices will be sparse but how is the claim in Theorem 1.1 correct?
- Writing could be a bit better by introducing Tucker Decomposition in the introduction. I found it a bit hard to follow without knowing what the core tensor and factor matrices are. This is probably not that big an issue as the target audience are aware of the terminology.

Some typos:
- In definition 3.2, you also need to state that $\||p\||_1 = 1$ -- not a big problem
- Missing $\||\cdot\||_2$ at a few places like in Corollary 3.5

———————————————————————————————————————
Update after author responses:
- Authors answered my questions and made a few changes to the manuscript which address most of my concerns.
- Although the paper does not contribute to the sketching literature as I said earlier, there are a lot of details and new techniques required to apply the existing techniques to the important problem of Tucker Decomposition which lead to my acceptance decision.

---

> ### Author Response · Authors · 2022-08-02
> **Author Response (Part II)**
>
> > If I understand correctly, the main speedup over [16] is obtained by not computing the product
> $\tilde{K}^\intercal \times \tilde{K}$ and instead using the Richardson's iteration which using `FastKroneckerProduct` you can perform quickly? Can you explain how the term $\tilde{O}(MM(\prod_{n \in T}R_n, R\varepsilon^{-1}, \prod_{n \not\in T}R_n))$ arises in line 13?
>
> Correct. Here’s what happens under the hood (Appendix B.1): We perform each Kronecker matrix-vector product $(A \otimes B)v$ as $(B \textnormal{mat}(v) A^T)$, where $\textnormal{mat}(v)$ is an appropriate matricization of vector $v$ (Lemma B.1). This means we can divide the matrices in the Kronecker product to two parts (in a balanced way) and multiply those from left and right with a matricization of our vector. Lastly, observe that we can permute the Kronecker factors at the beginning of the solve, which is why we optimize over the subsets of indices $[N]$.
>
> However, this process is not that straightforward in Theorem 4.5 because we are dealing with sampled vectors and matrices, so we need to be more careful. To the best of our knowledge, this sparse and fast Kronecker matrix-vector multiplication is a novel technique.
>
> Re main speedup: One of the key parts of the speedup comes from the fact that we do not compute $\tilde{K}^T \tilde{K}$ and its inverse. However, this can at most improve the running time to quadratic if we naively compute $\tilde{K} x$. To achieve subquadratic time, we perform Kronecker matrix-vector products in the novel way described above (Theorem 4.5). Furthermore, to get speedups for ridge regression and Tucker ALS factor matrix updates, we also need to use (approximate) SVDs and the Kronecker structure of $V$ judiciously. See the paragraph after Theorem 4.6 (Lines 285–295).
>
> > How does solving intermediate problems of ALS to a small approximation affect the overall convergence? I'd think it shouldn't be affected too much but do you have any small experiments where you observe anything significant?
>
> The convergence path of sketching-based Tucker ALS is very stable, even for large values of $\varepsilon$. Per your suggestion, we added Table 7 and Table 8 in Appendix D.2 to track the reconstruction errors at each step of ALS (with and without sketching) for decreasing values of $\varepsilon$. These results are for the cardiac MRI tensor, but we observed the same behavior for other image tensors and for synthetic data (e.g., compressing a tensor that has a known Tucker decomposition).
>
> > Some typos
>
> Thank you for catching these. Done.

---

> > ### Comment · Reviewer_yEQL · 2022-08-02
> > **Thanks for the rebuttal**
> >
> > Thank you for your detailed rebuttal and an updated version of the paper! I'll soon go over the new version of the paper and add my comments/update my review.

---

> > ### Comment · Reviewer_yEQL · 2022-08-03
> > **Some questions.**
> >
> > In Appendix C, you replace the condition of  $Nz = 0$ by adding a term $w \|\|Nz\|\|^2$ to the objective for a large enough $w$ and then do unconstrained optimization. How large of a $w$ do you choose here and how good is the solution $M^{+}z^*$ for the original problem if $z^*$ is a $1+\varepsilon$ approximate solution to the unconstrained problem obtained by adding $w$?
> >
> > Some suggestions:
> > 1. In the final version, maybe include the table which compares the running times in the Appendix in the introduction section and stress that you want to achieve running times that do not depend on $I_1 I_2 \cdots I_n$ which is why you focus more on the sampling algorithms which would lead to algorithms that look at only few entries of the tensor in each iteration.
> >
> > 2. In theorem 1.2, you skipped explaining what $k$ is. I assume that the running time is when $A_k$ is the factor matrix that is being updated?
> >
> > Rest of the paper looks good. Thanks for addressing my questions/concerns! I'll update my main review soon.

---

> > > ### Author Response · Authors · 2022-08-04
> > > **Author Response (Part III)**
> > >
> > > Thank you for the second round of feedback!
> > >
> > > > ​​In Appendix C, you replace the condition of $Nz = 0$ by adding a term $w ||Nz||^2$ to the objective for a large enough $w$ and then do unconstrained optimization. How large of a $w$ do you choose here and how good is the solution $M^{+}z^*$ for the original problem if $z^*$ is a $1+\varepsilon$ approximate solution to the unconstrained problem obtained by adding $w$?
> > >
> > > We added Lemma C.1 in Appendix C to show how large $w$ needs to safely solve the unconstrained problem. For the $n$-th factor matrices, it follows that
> > > $w = \Omega\left( \frac{1}{\varepsilon} \left\Vert \begin{bmatrix}
> > >   K \\\\ \sqrt{\lambda} (G_{(n)}^\intercal)^+
> > > \end{bmatrix} N^+ \right\Vert_2^2\right)$ suffices (Line 6 of Algorithm 3).
> > >
> > > > In the final version, maybe include the table which compares the running times in the Appendix in the introduction section and stress that you want to achieve running times that do not depend on $I_1 I_2 \cdots I_n$ which is why you focus more on the sampling algorithms which would lead to algorithms that look at only few entries of the tensor in each iteration.
> > >
> > > Excellent suggestion. We’ll move this table back to the introduction. (We initially cut it due to space.)
> > >
> > > > In theorem 1.2, you skipped explaining what $k$ is. I assume that the running time is when $A_k$ is the factor matrix that is being updated?
> > >
> > > Thanks for catching this. Yes, this is for updating factor matrix $A^{(k)}$. This is now explicit in Theorem 1.2.

---

> ### Author Response · Authors · 2022-08-02
> **Author Response (Part I)**
>
> Thank you very much for the constructive feedback. Please see our inlined responses.
>
> > The paper doesn't have any new contributions for sketching literature. Note that sketching a submatrix $A_1$ by leverage score sampling rows of $A_1$ using leverage scores with respect to $A$ obviously works because of the property that leverage scores only decrease when the new rows are added. This observation is not new and can be seen in the paper "Sharper Bounds for Regularized Data Fitting".
>
> While we agree we do not make any new contributions to the sketching literature, we want to emphasize that the goal of this paper is to *use sketching* to solve Kronecker regression faster and speed up Tucker ALS. Our work accomplished this goal both in theory and in practice.
>
> We were unaware of “Sharper bounds for regularized data fitting” (RANDOM 2017), but we added a citation for Theorem 2 (ridge regression) and updated the paragraph after Corollary 3.5 to better reflect known results.
>
> > Can you clarify my doubts about the input sparsity claims in Theorem 1.1?
>
> Yes, please see our updated Algorithm 2 and Theorem 1.1 in the revision. We have tried to make our ideas clear and concrete.
>
> In short, the algorithm as stated didn’t run in input-sparsity time due to exact SVDs as you pointed out (thanks for catching this), but the core idea can be lightly modified to go through. In Tucker ALS, we almost always deal with dense matrices and low multilinear rank (i.e., values of $R_i$), so computing the exact SVD for each factor Gram matrix is quite effective in practice as you observed.
>
> Re tightening up the input-sparsity claim: Note that we do not need to find the SVD of $A_i$, but rather we need to compute the SVD of $A_i^T A_i$. This is an $R_i \times R_i$ matrix. Moreover, instead of computing the SVD of $A_i^T A_i$, we can find a spectral approximation of $A_i^T A_i$ (e.g., by leverage score sampling in time $O(nnz(A)+R^\omega \varepsilon^{-2})$ using [Lemma 4, Cohen et al., ITCS 2015]). So suppose for all $A_i$, we find $\tilde{O}(R_i \varepsilon^{-1}) \times R_i$ matrices $B_i$ such that $A_i^T A_i \preceq B_i^T B_i \preceq (1+\varepsilon’) A_i^T A_i$. Then setting $\varepsilon’ < \log(1+\varepsilon)/N$, we have $A^T A \preceq B^T B \preceq (1+\varepsilon) A^T A$, where $A$ is the Kronecker product of the $A_i$’s and $B$ is the Kronecker product of the $B_i$’s. Then we can compute $B_i^T B_i$ in $\tilde{O}(R_i^\omega \varepsilon^{-2})$ time and compute the SVD of $B_i^T B_i$ in $O(R_i^\omega \varepsilon^{-2})$ time. Our algorithm works similarly to this approach and gives input-sparsity time. We explain this in more detail in the paragraph after Theorem 4.6 (Lines 285–295) in the revised version of the paper. We also added more details to Algorithm 2 and the proof of Theorem 4.6 in Appendix B.2.
>
> > I am confused about where Theorem 4.5 is used in `FastKroneckerRegession`. Isn't the only product of a vector with a Kronecker product in line 13? And the vector there isn't sparse. Am I missing anything in regards to where it is used?
>
> Yes, Theorem 4.5 is only used in Line 14 of Algorithm 2. There are five Kronecker matrix-vector products per update to $x$. Note that all of the dimension-$I$ intermediate vectors are actually sparse with $\tilde{O}(R \varepsilon^{-1})$ nonzeros, so we can compute them efficiently by Theorem 4.5:
> $\tilde{K}^T \tilde{b} = (SK)^T Sb$
> $\tilde{K} x = S K x$

---

### Official Review · Reviewer_Ubgq · 2022-07-10

**Rating:** 7
**Confidence:** 3
**Soundness:** 3 good
**Presentation:** 3 good
**Contribution:** 3 good

**Summary:**

The authors propose the first subquadratic-time algorithm for solving Kronecker regression. Kronecker regression is a structured problem where the design matrix is a Kronecker product of multiple matrices. This problem appears when trying to compute the Tucker decomposition of a Tensor. The use score sampling and iterative methods to achieve a complexity reduction for Kronecker regression problems. Then, they extend their approach to block design matrices. They use synthetic and real examples to demonstrate the computational advantages of their approach.

**Questions:**

 Can you also provide examples on real data?
What is nnz on page two? This is now explained.
Perhaps the paper would read better if the theorems can be pushed forward outside of the introduction.
P5L201 what is s (the scalar s).


**Limitations:**

The limitations of the proposed work are properly detailed in section F.

**Strengths And Weaknesses:**

The paper reads well, and the English level is satisfactory. I am not familiar with how important the addressed problem is, but the method seems novel with solid justification, both theoretical and empirical. The runtime improvement seems substantial, and the relative error of the algorithm on synthetic data seems low. The related works are reviewed well and proper background to the problem is provided.

---

> ### Author Response · Authors · 2022-08-02
> **Author Reponse**
>
> We appreciate your careful review and suggestions. Please see our inlined responses.
>
> >  I am not familiar with how important the addressed problem is, but the method seems novel with solid justification, both theoretical and empirical.
>
> The Tucker decomposition is the second-most used/studied tensor decomposition after the CP decomposition. Alternating least squares (ALS) is the main algorithm for computing CP and Tucker decompositions in practice (e.g., see the algorithms listed in MATLAB’s Tensor Toolbox). We speed up all steps of this algorithm in both theory and in practice for Tucker decompositions with sufficiently large core tensors.
>
> > Can you also provide examples on real data?
>
> We used our `FastKroneckerRegression` algorithm as a subroutine in Tucker ALS to compress three benchmark image tensors in Appendix D (e.g., a 4-dimensional cardiac MRI). We compare against the exact HOOI Tucker decomposition algorithm and Diao et al. [16] for approximate Kronecker regression, and we report running times and relative reconstruction errors.
>
> > Perhaps the paper would read better if the theorems can be pushed forward outside of the introduction.
>
> That would certainly make the paper feel less rushed, but the convention for theory-heavy conference papers is to make the main formal results accessible as soon as possible.
>
> > P5L201 what is s (the scalar s).
>
> The scalar $s$ is the number of row samples. We will try to make that more clear.

---

> > ### Comment · Reviewer_Ubgq · 2022-08-08
> > **Response to authors**
> >
> > Thanks for addressing all my comments; I keep my score at acceptance; good luck!

---

### Official Review · Reviewer_FzUX · 2022-07-11

**Rating:** 7
**Confidence:** 2
**Soundness:** 4 excellent
**Presentation:** 3 good
**Contribution:** 4 excellent

**Summary:**

The work considered the fundamental problem of Kronecker regression and proposed the first sketching-based subquadratic algorithm to obtain a (1+\epsilon)-approximation solution. Both rigorous proofs and empirical validations were shown. The block-sketching methodology can be of independent interests itself.

**Questions:**

In the experiment part, was fast matrix multiplication methods used? How does the runtime complexity compared to the theoretical results in Theorem 1?

**Limitations:**

Yes.

**Strengths And Weaknesses:**

The work focused on an accelerated solution for a fundamental problem which makes the work very meaningful. The technique itself is quite interesting. The math is well presented and validated by empirical results.

---

> ### Author Response · Authors · 2022-08-02
> **Author Response**
>
> Thank you for your positive review and questions. Please see our inlined responses.
>
> > In the experiment part, were fast matrix multiplication methods used?
>
> We used `numpy.matmul` for matrix multiplication, which calls well-optimized C code under the hood. The theoretically interesting algorithms that run in time $O(n^{\omega})$ are highly impractical because of their massive constant coefficients hidden by big O notation.
>
> That said, we used our implementation of `KronMatMul` (Lemma 4.4) for the Kronecker matrix-vector products to drop the running times from $\tilde{O}(R^3 \varepsilon^{-1})$ to $\tilde{O}(R^2 \epsilon^{-1})$ in practice.
>
> > How does the runtime complexity compare to the theoretical results in Theorem 1?
>
> We computed the SVD of the Gram matrix of each factor to get exact leverage scores (Line 4 of Algorithm 2), and we used our `KronMatMul` instead of the balanced version based on theoretically-fast rectangular matrix multiplications. Therefore, the running time in the experiments is $O(\sum_{n=1}^N (I_n R_n^2 + R_n^3) + R^2 \varepsilon^{-1})$ compared to Theorem 1.1.

---

### Official Review · Reviewer_djSP · 2022-07-20

**Rating:** 7
**Confidence:** 3
**Soundness:** 4 excellent
**Presentation:** 4 excellent
**Contribution:** 3 good

**Summary:**

This paper provides faster algorithms for solving the $\ell_2$ - Kronecker product regression problem, which is a least squares regression problem where the design matrix is a Kronecker product of several smaller matrices with dimensions $I_n \times R_n, n \in [N]$. They also extend their techniques to provide faster algorithms for Kronecker ridge regression, and speed up the Alternating Least Squares (ALS) algorithm for computing the Tucker decomposition of a tensor. Prior work (Diao et.al, 2019) for Kronecker product regression cared about the case where $R_n \gg I_n$ for all $n \in [N]$ and this work improves the runtime when $R_n$ might be comparable to $I_n$ and justify that this case is important through their experiments.

Reference:
Huaian Diao, Rajesh Jayaram, Zhao Song, Wen Sun, and David Woodruff. Optimal sketching for kronecker product regression and low rank approximation. Advances in neural information processing systems, 32, 2019.

**Questions:**

I briefly went over the Diao et.al [16] paper and their runtimes seem to have a factor of $1/\epsilon^2$ (see footnote 9 on page 6) rather than $1/\epsilon$ as claimed on Line 46 of this paper?

Editing suggestions:
It would be better if you define $\omega$ (along with a reference and an approximate value) before using it on page 2.

Line 93, 260 - “pseduoinverse” -> “pseudoinverse”



**Limitations:**

Primarily theoretical work.

**Strengths And Weaknesses:**

Originality: I am not an expert in this area so I am not entirely sure about other related work and how novel this work's contributions are compared to other papers in the area.

Quality: The submission is technically sound. All claims are well-supported with proofs.

Clarity: The submission is clearly written and well-organized.

Significance: Yes, the problems being studied are significant. Kronecker product regression is a well-studied problem with several applications.

---

> ### Author Response · Authors · 2022-08-02
> **Author Response**
>
> Thank you for your thorough review. Please see our inlined responses.
>
> > I briefly went over the Diao et. al [16] paper and their runtimes seem to have a factor of $1/\varepsilon^2$ (see footnote 9 on page 6) rather than as claimed on Line 46 of this paper?
>
> True. Their $p=2$ results (Algorithm 1 and Theorem 3.1) are stated using $m = \theta(d / (\delta \varepsilon^2))$ samples.
>
> This is a subtle point because only $m = \theta(d / (\delta \varepsilon))$ samples are needed in their analysis. Specifically, they apply [Lemma 3.3, CW09] to prove Theorem 3.1, but this only requires $m = \theta(d \log(1/delta) / \varepsilon)$ samples. The increased number of samples is first introduced in Proposition 3.4 to compute high-accuracy approximate leverage scores for each factor matrix $A_i$, but then $(1 \pm 1/q)$-approximations are used in Proposition 3.5 (much weaker). This refined analysis agrees with our results, as well as the sample complexity in “Active regression via linear-sample sparsification” [Chen and Price, COLT 2019].
>
> Fundamental reason for this: The dependency on $\varepsilon$ can be reduced from $O(\varepsilon^{-2})$ to $O(\varepsilon^{-1})$ in approximate regression because it suffices for the sketch to be a $(1 \pm 1/\sqrt{2})$-spectral approximation, not a stronger $(1 \pm \varepsilon)$-spectral approximation. See the “Satisfying structural condition 1” paragraph in our proof of Lemma 3.4 in Appendix A for the full details.
>
> > It would be better if you define  (along with a reference and an approximate value) before using it on page 2.
>
> Done.
>
> > Line 93, 260 - “pseduoinverse” -> “pseudoinverse”
>
> Thanks for catching this. Fixed.

---

### Author Response · Authors · 2022-08-02
**Revision Changes**

We would like to inform the reviewers that we uploaded a revised version of our paper based on your comments and suggestions. Here is a list of the changes:
- Modified Algorithm 2 and the proof of Theorem 4.6 in Appendix B.2 to accurately reflect the input-sparsity claim in Theorem 1.1 in the initial submission (i.e., now we compute approximate SVDs for the factor Gram matrices in input-sparsity time instead of exact SVDs). We also updated the algorithm description in Section 4.3 to better describe this.
- Added Table 7 and Table 8 in Appendix D.2 to illustrate the stability of the convergence paths in sketching-based Tucker ALS for different values of the error parameter $\varepsilon$.
- A small modification to the first bullet in Section 1.1 to better describe our techniques.
- Cited "Sharper bounds for regularized data fitting" (Avron–Clarkson–Woodruff, RANDOM 2017) and mentioned its contribution to approximate ridge regression and block sketching.
- Fixed all minor typos and comments.

---

### Meta-Review · Area_Chair_onCi · 2022-08-24

**Recommendation:** Accept
**Confidence:** Less certain

**Metareview:**

All reviewers felt that this paper made a solid technical contribution on algorithms for Kronecker regression and should be accepted to the conference.


**Award:**

No

---

### Decision · Program_Chairs · 2022-09-14

Accept